# An Explorative Model to Assess Individuals' Phubbing Risk

**Andrea Guazzini** [1,*], **Mirko Duradoni** [2], **Ambra Capelli** [1] **and Patrizia Meringolo** [1]

[1]  Department of Education and Psychology, University of Florence, and Center for the Study of Complex Dynamics (CSDC), Via di San Salvi 12, 50135 Florence, Italy; mildram06@gmail.com (A.C.); patrizia.meringolo@unifi.it (P.M.)

[2]  Department of Information Engineering, University of Florence, via S. Marta 3, 50139 Florence, Italy; mirko.duradoni@unifi.it

*   Correspondence: andrea.guazzini@unifi.it

**Abstract:** Phubbing could be defined as a new form of addiction; however, checking the phone and ignoring the speaker could also be linked to the increased availability of virtual social environments. We developed a multidimensional model for phubbing considering psychological dimensions and information and communication technology related habits. We collected data through online questionnaires and surveys. The best model obtained from our data was constituted by Information and Communication Technologies' (ICTs) usage behaviours, Trait Anxiety, Virtual Sense of Community and Neuroticism. Finally, our study confirmed a strong connection between phubbing and online addiction behaviours.

**Keywords:** phubbing; cyberpsychology; internet addiction; smartphone addiction

## 1. Introduction

The impact of Information and Communication Technologies (ICTs) (e.g., smartphone, tablet, kindle, smartwatch) on people's everyday lives is very noticeable [1,2]. Indeed, for instance, millions of people all over the world relate to others simply by using their phone [3]. Despite merely remaining updated about who contacted us, smartphones and other ICTs devices also permit handling our own virtual social identity and keep in touch with relevant virtual social communities [4]. In such sense, virtual environments have a strong social attractiveness. Consequently, brand new psychological questions and issues emerged. Indeed, while the world is becoming more and more connected, people can develop addictions in response to this wider possibility of virtual contacts and thus becoming more disconnected from reality [5].

Despite recent studies that highlighted a positive influence of smartphones in professional environments such as health care coordination [6], infrastructure monitoring [7], and in promoting socialization with geographically distant individuals, in other cases, smartphone usage could be detrimental for individuals' well-being [8–10].

Phubbing behavior (i.e., the habit of snubbing someone in favor of a mobile phone) has recently received a growing attention among those psychological issues and consequences related to smartphone usage [11–14]. Phubbing is derived from the union of the words "phone" and "snubbing", and describes the action of ignoring someone in a social environment by looking at the phone instead of paying attention to the other person [15]. Phubbing is also prominent during intimate social interactions. For instance, a large number of couples interrupt repeatedly their meal while eating together to check their phone for messages or missed calls [16].

Phubbing is considered by the scientific literature as a new form of addiction [14,17], a compulsive behavior realized in order to temporary escape and avoid a particular stressful situation or negative thoughts and emotions. Since more and more people are becoming addicted to their smartphones [18], web-based platform and online services relying on mobile phones should rapidly find a solution to avoid addiction-related taxes (similarly to cigarette use) and to meet well-being required standards [19]. One possible way to achieve this goal relies on understanding which individual characteristics are linked to smartphone addiction and thus profiling the user according to these parameters to assess their phubbing-related risk. However, the current literature is a little meagre, and there are only a few studies that investigate phubbing's possible predictors and they certainly did not give us the full picture of this complex phenomenon. For instance, psychological constructs like anxiety, self-efficacy and personality were not fully taken into consideration and several studies demonstrated that all these observables can trigger a compulsive behavior [20–22]. In particular, neuroticism, trait anxiety, as well as social involvement, appeared very related to online addiction behaviors [23,24].

On the one hand, phubbing could be defined as a form of addiction, in which the compulsive component appears to be preeminent. On the other, checking the phone ignoring the speaker could also be linked to the increased availability of virtual social environments. In the latter case, the attention could simply be directed towards the social group perceived at that moment as more salient. Smartphones and virtual environments' great availability potentially increased the number of social identities to be managed at the same time. For instance, people often use more than a social network [25]. Furthermore, people try to re-create their offline self online. However, individuals spend efforts in editing self's facets to project a given identity online [26]. Thus, virtual identities managing process could be very time consuming and can lead to privilege virtual environments even to the detriment of face-to-face interactions. Phubbing could also be related to selfishness. In other words, phubbers could prefer their own online selves over the social interaction with another individual. Since males are reported in the scientific literature as more selfish than females, gender could affect phubbing dynamics [27,28]. Clearly, phubbing could be determined by both an online identity management process and an avoidance necessity. The analysis of the interactions between these dynamics will allow us to build a first and exploratory multivariate model to assess phubbing risk extending the previous work of Guazzini, Capelli and Meringolo [29].

## 2. Aims of the Study

Overall, our work aims to explore the determinants of phubbing (ranging from a mere interruption of the face-to-face interaction to a phone obsession) developing a multidimensional model considering all dimensions that the scientific literature has shown to be related to this behavior (e.g., sociodemographic and psychological variables). Given the previous works, several hypotheses have been formulated.

First, we expect phubbing to be positively related with phone-related addiction (e.g., use of Short Message Service, SMS) and with Internet and social media addiction [11,12]. Indeed, in the first case, the phone is a medium through which addiction is substantiated, while surfing on the Internet and social network attendance could be thought as the objects to which addiction is directed.

We also expect that the ICT pervasivity measure would be associated with phubbing. Indeed, the more connected devices people have, the more the possibility for them to engage in phubbing [3]. However, we hypothesize that the simple number of social networks used by a person will not necessarily imply a need to check the phone, while it could affect the communication disturbance component of phubbing [5]. Again, the more social networks a person uses, the greater the occasions in which could be reached by a notification or be incentivized to check the phone during a face-to-face interaction.

Finally, we expect that psychological dimension like Neuroticism, Trait Anxiety, as well as Virtual Sense of Community could affect the phubbing obsession component [20–24].

## 3. Sampling and Participants

The research has been conducted on a sample of 394 individuals. The data obtained from some people, who gave the same score to all items and who were believed to have responded dishonestly, affecting the validity of this research negatively, were removed before the analysis. Thus, data from 361 participants were used in the research. The sample responded to an online questionnaire, design ad hoc, in total anonymity. Since the real identity of the subjects wasn't collected, we didn't proceed with asking for informed consent. The sample was recruited through personal contact and online posts on the major Social Network Sites. All of the participants were volunteers. Of the participants, 306 were female (84.8%) and 55 male (15.2%) with an age from 15 until 68 (M: 24,16; SD: 8,14). The sample turn out to be composed for the majority of Italian people (98.6%), except for five individuals (1,4%), and most of them were full-time students (71.7%). In addition, 98.1% of the participants owns a smartphone.

## 4. Methods and Procedures

The data were collected thanks to the use of Google modules that allowed us to create an online version of our questionnaire and to easily send it through email and social network sites. We decided to proceed with an online administration, rather than face-to-face, because, from literature, it has been found that, online, people are more inclined to give sensitive information (i.e., more real self-disclosure) and to give more honest answers [30]. The final questionnaire asked for data of the sociodemographic background, of the ICTs and Social Network Sites usage, and used different scales to investigate the phubbing behavior and personal characteristic. For most of the scales, we used the validated Italian version but for a few of them, like the phubbing and partner phubbing scales, not available in Italian, we proceed with a forward and back translation. The instruments that we used are:

*The phubbing scale* [12] consists of 10 items graded from 1 (never) to 5 (always) on a 5-point Likert scale divided into two factors:

- *Communication disturbances* (5 items; $\alpha$ = 0.87): high scores indicate that the person often disturbs the communication using the smartphone in a face-to-face environment. Examples of this factor's items are: "My eyes go to the phone when I'm together with others" and "I'm dealing with my mobile phone when I'm with my friends".
- *Phone Obsession* (5 items; $\alpha$ = 0.85): high scores indicate that the person feels the constant need of his/her smartphone in an environment where there's a lack of a face-to-face communications. Examples of this factor's items are: "My phone is always within my reach" and "When I wake up in the morning, I first check my messages on my phone".

The *Partner Phubbing Scale* [15] investigates the extension of the smartphone usage when someone is in company of his/her own partner. It consists of nine items, graded from "Never" (1) to "Sometimes" (3) to "All of the time" (5) with a reliability of 0.93. Examples of items are: "My partner places his or her cell phone where they can see it when we are together" and "My partner keeps his or her cell phone in their hand when he or she is with me".

The *Mobile Phone Usage Addiction Scale* [12] evaluates the mobile phone usage addiction and consists of 15 items, graded from 1 (never) to 5 (always) on a 5-point Likert scale. The 15 items of this scale form three dimensions: deprivation (7 items, $\alpha$ = 0.86) (e.g., I feel anxious when I don't have my mobile phone with me), control difficulties (3 items, $\alpha$ = 0.78) (e.g., I had times when I was so busy with the mobile phone and I lost the track of time), and application (5 items, $\alpha$ = 0.85) (e.g., the applications in my mobile phone make my daily works easier).

The *SMS Addiction Scale* [12] investigates the extent of the SMS addiction. This scale consisted of six items graded from 1 (never) to 5 (always) on a 5-point Likert scale. Examples of items are "I feel a need to reply the messages instantly" and "I keep online the messaging applications all the time". All of the items load in only one factor and the Cronbach's alpha coefficient for the SMS Addiction Scale was 0.80.

The *Game Addiction Scale* [12] is used to establish the addiction to games and consists of eight items (e.g., I lose track of time when I play games; delay sleeping hours when I play a game) graded from 1 (never) to 5 (always) on a 5-point Likert scale. All the items load in only one factor and the Cronbach's alpha coefficient for the Game Addiction Scale was 0.90.

The *Social Media Addiction Scale* [12] was developed to investigate the addiction specifically to social media. This scale consisted of 10 items graded from 1 (never) to 5 (always) on a 5-point Likert scale and it has a two-factors structure. The two factors are sharing (6 items, $\alpha$ = 0.82) (e.g., I share what I did, what is going on with life and momentary events in social media) and control (4 items, $\alpha$ = 0.79) (e.g., I check over my social media accounts whenever possible).

The *Internet Addiction Scale* [12] measures the internet addiction. This scale consisted of six items (e.g., I spend time using the Internet more than I plan to) graded from 1 (never) to 5 (always) on a 5-point Likert scale. All the items load in only one factor and the Cronbach's alpha coefficient for the Internet addiction scale was 0.83.

We investigated the ICTs' pervasivity asking in which contexts and environments the participants use the online services giving the possibility of multiple choice. We chose as indicators contexts like *at school/university*, *in the free time*, *with family*, *with friends*, *in case of emergency*, *while shopping* and *at work*. The more ICTs contexts are selected, the more participants' ICTs pervasivity results in being high.

In order to investigate the personality, we used the *I-TIPI Scale* in the validated Italian version [31] developed from the original scale of [32].The scale has 10 items graded from Disagree strongly (1) to Agree strongly (7) on a 7-point Likert scale. The items form five dimensions: Extraversion (e.g., Extraverted, enthusiastic); Agreeableness (e.g., Sympathetic, warm); Conscientiousness (e.g., Dependable, self-disciplined); Emotional Stability (e.g., Calm, emotionally stable); Openness to Experiences (e.g., Open to new experiences, complex).

To measure anxiety, we used the *Stai scale* of [33] in its Italian version [34]. STAI is a psychological inventory that consists of 40 self-report questions, divided in two scales that focus on how people feel generally (trait anxiety—20 items) or on how they feel in that particular moment (state anxiety—20 items). In our questionnaire, we used only the trait anxiety scale. The 20 items are graded from 1 (almost never) to 4 (almost always) on a 4-point Likert scale (e.g., I feel nervous and restless; I have disturbing thoughts; I am a steady person). Internal consistency coefficients for the scale have ranged from 0.86 to 0.95.

The *Self-Efficacy Scale* [35] (here in the Italian version [36]) was used to investigate the perception of self-efficacy. The scale consists of 10 items graded on a 4-point Likert scale from "Not at all true" (1) to "Exactly true" (4). Exemples of items are: "I can always manage to solve difficult problems if I try hard enough", "I am confident that I could deal efficiently with unexpected events". Cronbach's alphas ranged from 0.76 to 0.90. The scale is unidimensional.

The *Self-Esteem Scale* [37] measured the self-esteem of one person considering the positive and negative feelings towards one's self. The scale has 10 items (e.g., on the whole, I am satisfied with myself; I feel that I'm a person of worth) measured on a 4-point Likert scale graded from 1 (Strongly agree) to 4 (Strongly disagree) The reliability analysis yielded an alpha coefficient of 0.86 with item correlation coefficients ranging from 0.25 to 0.76, indicating good internal reliability.

The *Sense of Virtual Community Scale* [38] investigates the sense of virtual community, meaning the sense of belonging and attachment that one person feels towards a community, in this case in virtual environment. The scale has 18 items graded from 1 (strongly disagree) to 4 (strongly agree) on a 4-point Likert scale. Examples of items: "think this group is a good place for me to be a member", "If there is a problem in this group, there are members here who can solve it". The internal reliability coefficient for the scale is 0.93.

The *Perceived Social Self-Efficacy Scale* [39]. The scale contains 25 items (e.g., "Put yourself in a new and different social situation" and "Find someone to go to lunch with") on a 5-point Likert-type scale (1 = no confidence at all to 5 = complete confidence). The scale items are related to making friends, social

assertiveness, pursuing romantic relationships, performance in public situations, groups and parties, and receiving and giving help. This scale has a single-factor structure with coefficient alpha = 0.94.

The *Social Interaction Anxiety Scale* [40] investigates the distress that individuals experience when meet or talk to other people. It consists of 20 items (e.g., When mixing socially, I am uncomfortable; I am tense mixing in a group) on a 5-point Likert scale graded from "not at all" (0) to "extremely" (4). the Cronbach's alpha coefficient for the Social Interaction Anxiety Scale is 0.93.

## 5. Data Analysis

The statistical procedures adopted to treat and analyse the data have been divided along three phases. In the first phase, the data were collected, cleaned, and the outliers were eliminated. Therefore, the preliminary conditions required by the inferential analysis planned by the study were verified (i.e., minimal sample sizes, balance and normality of continuous variable distributions, by means of skewness and kurtosis). In the second phase, the descriptive statistics were produced, and, in the third phase, the inferential analysis was carried out. In order to answer to main hypotheses of the paper, we adopted the Pearson *r.* correlation to explore the relation between continuous variables, and the multiple linear regression modeling in order to model the multiple effects acting on the Social Media usage (i.e., Social Media Addiction Scale score). Finally, a multivariate analysis of variance and covariance (MANCOVA) has been adopted to develop the best model explaining the phubbing network of relations with the observables considered by our study.

## 6. Results

The results of the study are organized along four subsections. Each subsection answers a subset of preliminary hyphotheses regarding the same "family" of theorethical constructs and relations. In the first section, the descriptive statistics are provided for all the variables considered by the study. In the second section, we provided the univariate statistics describing the complex network of relations between the operative variables (i.e., phubbing factors) and the ICTs' usage dimensions, as well as the psychological dimensions. In the third section, a multivariate modeling of Social Media usage is presented, and, in the fourth and last section, a multivariate analysis of variance and covariance (MANCOVA) is provided to understand the complex network of relations associated with the phubbing phenomena.

### 6.1. Descriptive Statistics

Descriptive statistics of the psychological and digital life dimensions considered in our study are shown in Table 1. These dimensions, for which we reported the average with standard error and standard deviation, skewness and kurtosis in the table below, represents in general how our subjects responded to our questionnaire and the data show the normality of distribution.

**Table 1.** Descriptive statistics of psychological and digital life dimensions.

| Descriptive Statistics | | | | |
|---|---|---|---|---|
| **Psychological Dimensions** | | | | |
| **Variable** | **Average (SE)** | **Std. Dev.** | **Skewness** | **Kurtosis** |
| Neuroticism | 6.12(0.11) | 8.11 | −0.17 | −0.70 |
| Trait Anxiety (STAI) | 47.78(0.56) | 10.61 | 0.24 | −0.02 |
| Sense of Virtual Community (SVC) | 27.01(0.40) | 7.58 | 0.15 | 0.05 |
| General Self Efficacy (GSE) | 28.17(0.28) | 5.38 | −0.09 | 0.01 |
| Social Anxiety (SIAS) | 48.09(0.80) | 15.19 | 0.38 | −0.35 |
| **Digital Life Dimensions** | | | | |
| **Variable** | **Average (SE)** | **Std. Dev.** | **Skewness** | **Kurtosis** |
| Mobile Phone Usage Scale | 40.15(0.49) | 9.33 | 0.27 | 0.45 |
| SMS Usage Scale | 15.10(0.21) | 4.07 | 0.26 | 0.43 |
| Games Usage Scale | 13.17(0.29) | 5.63 | 1.14 | 0.54 |
| Social Media Usage Scale | 26.81(0.39) | 7.41 | 0.12 | 0.16 |
| Internet Usage Scale | 13.51(0.27) | 5.18 | 0.70 | −0.14 |
| ICT Usage frequency [1] | 4.78(0.03) | 0.50 | −0.35 | 0.53 |
| Number of ICT Services owned | 5.29(0.08) | 1.61 | 0.05 | −0.10 |
| ICT Social Pervasiveness | 3.96(0.78) | 1.47 | 0.23 | −0.57 |
| Number of SNSs | 3.31(0.07) | 1.37 | 0.72 | 1.22 |
| SNSs daily accesses [2] | 2.53(0.04) | 0.76 | 0.56 | 0.78 |
| SNSs daily duration of connections [3] | 2.17(0.05) | 0.93 | 0.53 | −0.07 |
| Number of Activities on SNSs | 3.35(0.08) | 1.54 | 0.96 | 1.15 |
| Number of Topics on SNSs | 3.57(0.10) | 1.97 | 0.62 | −0.04 |
| Frequency of contacts on SNSs [4] | 2.63(0.05) | 0.92 | 0.64 | 0.26 |

[1]: ICT Usage frequency average is between often (one time a day) and always (more than one hour a day). [2]: SNSs daily accesses between 1 and 50 times a day. [3]: SNSs daily duration of connection is between 1 and 4 h a day. [4]: Frequency of contacts on SNSs is around rarely and sometimes.

Table 1 reports the descriptive statistics and it shows that the subjects who responded to our questionnaire use the ICT (predominantly smartphone and computer) at least one time a day for more than one hour. They use them to access Social Network Sites (SNSs) more than one time a day for one to four hours. On average, each person owns at least three different social networks that they use for various activities like discussing different topics and sometimes being in touch with their contacts.

Table 2 illustrates the average with standard error and standard deviation, skewness and kurtosis for the operative variables. The experimental data reports, both for Personal Phubbing Scale and Partner Phubbing Scale, an average behaviour very close to those reported in literature. In particular, we found an average for the Personal Phubbing Scale of 2.74, in line with Karadag et al., which reported an average of 2.76 [12]. Concerning the Partner Phubbing Scale, the data reports an average of 2.54 accordingly with Roberts and David that, in their study, found an average of 2.64 [15].

All of the operative variables show an acceptable normal distribution (i.e., the values of skewness and kurtosis ranged between −1 + 1.

**Table 2.** Descriptive statistics of operative variables.

| Operative Descriptive Statistics | | | | |
|---|---|---|---|---|
| **Variable** | **Score (SE)** | **Average** | **Std. Dev.** | **Skewness** | **Kurtosis** |
|---|---|---|---|---|---|
| Personal Phubbing Scale (PePS) | 27.41(0.31) | 2.76 | 5.97 | 0.06 | −0.20 |
| PePS Factor: Communication Disturbances | 11.89(0.17) | 2.38 | 3.18 | 0.30 | 0.10 |
| PePS Factor: Phone Obsession | 15.52(0.19) | 3.10 | 3.58 | −0.12 | −0.43 |
| Partner Phubbing Scale (PaPS) | 22.91(0.35) | 2.64 | 6.67 | 0.31 | −0.22 |

*6.2. Result 1: Univariate Operative*

Before proceeding to our inferential analyses, we tested by means of a univariate analysis of variance whether gender could affect phubbing. In other words, gender has been introduced as a dummy factor [41]. No statistically significant difference emerged in relation to our dependent variables (i.e., Phubbing Factor: Communication Disturbance, Phubbing Factor: Phone Obsession, Personal Phubbing (Total), and Partner Phubbing).

6.2.1. Phubbing Univariate Predictors: Psychological and Sociodemographical Effects

Table 3 illustrates the correlation between the operative and the psychological and sociodemographic variables. As shown in the table, age and self-efficacy have a negative significant correlation with both personal phubbing and partner phubbing pointing out that younger people report to do and to suffer more from phubbing than elder people. Medium-low positive correlation emerge with psychological variables like anxiety, social anxiety and neuroticism.

**Table 3.** Pearson *r.* correlations between personal and partner phubbing and sociodemographic (age) and psychological variables, like anxiety and self-efficacy.

| Observable | Social Anxiety | STAI (Trait) | General Self Efficacy | Neuroticism | Age |
|---|---|---|---|---|---|
| Phubbing Factor: Communication disturbance | 0.282 *** | 0.281 *** | −0.183 *** | 0.233 *** | −0.251 *** |
| Phubbing Factor: Phone obsession | 0.160 *** | 0.157 *** | −0.112 * | 0.214 *** | −0.195 *** |
| Personal Phubbing (Total) | 0.246 *** | 0.244 *** | −0.165 *** | 0.252 *** | −0.250 *** |
| Partner Phubbing | ns | 0.151 ** | ns | ns | ns |

*** = $p$ 0.001, ** = $p$ 0.01, * = $p$ 0.05.

6.2.2. Phubbing Univariate Predictors: ICT and Social Media Effects

Overall, the social media effects (i.e., social media and ICT usage and related addiction scales) have been previously assessed by research. Following the literature, we explored the general correlation structure emerging from those observables of our study that fit with the literature evidence (Table 4). In particular, in accordance with literature, the Mobile Phone Usage appears to be the best predictor, within the ICT related features, of all the phubbing factors' scores, as well as of the partner phubbing score. A strong relation is evident also, as predictable, with the other scales. The only exception is represented by the Games Usage Addiction Scale (GUAs). Nevertheless, the strength of relations with the partner phubbing appear in general to be weaker.

**Table 4.** Pearson *r.* correlations between personal and partner phubbing and the ICT usage addiction.

| Observable | Phubbing Factor 1 | Phubbing Factor 2 | Total Phubbing | Partner Phubbing |
|---|---|---|---|---|
| Mobile Phone Usage | 0.524 *** | 0.597 *** | 0.636 *** | 0.294 *** |
| SMS Usage/Addiction | 0.441 *** | 0.460 *** | 0.510 *** | 0.228 *** |
| Games Usage/Addiction | 0.162 *** | 0.130 ** | 0.164 *** | 0.154 *** |
| Social Media Usage/Addiction | 0.499 *** | 0.498 *** | 0.564 *** | 0.293 *** |
| Internet Usage/Addiction | 0.493 *** | 0.413 *** | 0.510 *** | 0.254 *** |

*** = $p$ 0.001, ** = $p$ 0.01, * = $p$ 0.05.

Figure 1 is a recap of the results explained in details above. As we can see in this picture, there are strong positive correlations between both factors of phubbing and the addiction investigated in our study. The strongest one is with the mobile phone/smartphone usage addiction. A good relation is also present between phubbing and partner phubbing, highlighting how to do phubbing is correlated to our perception on how much one's partner does phubbing in his/her presence and thus on how much someone feels they are being phubbed. All correlations are reported in Table 4.

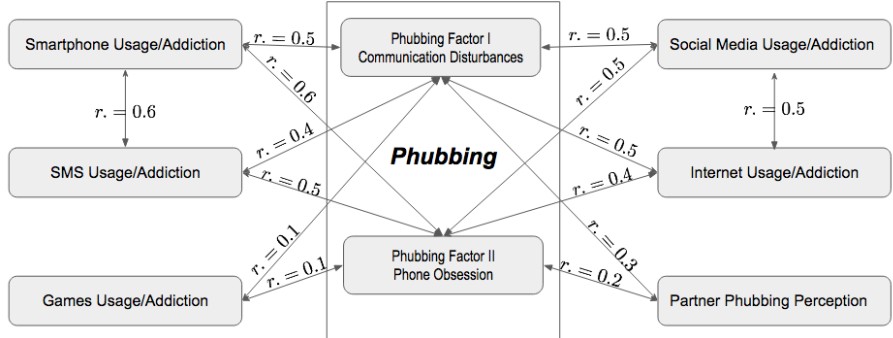

**Figure 1.** Summary of correlations between phubbing and ICT based predictors. Pearson *r.* correlation between phubbing and ICT usage addiction features.

### 6.2.3. Phubbing Multivariate Modeling

MANCOVA analysis allows us to study the effects of our factors of interest, on a centroid variable merging different and correlated observables representing the complexity of a certain phenomenon. In this way, it is possible to appreciate (i.e., estimate and validate) the single and combined effects of the model factors, on both the single dimensions composing the centroid i.e., Phubbing Factor 1 and also on the centroid itself (i.e., general model). In the upper part of Table 5, the best general model refined by the analysis is presented. The model explains 36% of the variance of the centroid, and there are eight factors maintained by the model. The power of the test is always greater than 0.7, and the $\eta^2$ tells the percentage of explained variance by every factor. The factor which explains the greater quantity of variance is Social Media Usage (i.e., 7.2%), with the SMS usage explaining the 5.4%, and the Internet Usage the 5.9%. A moderate effect ranging around the 3% is played by the ICT pervasivity, the number of SNSs owned, the Neuroticism, the Anxiety and the Virtual Sense of Community of the subject.

The principal effects, i.e., the effects of the factors on the single components of the centroid, are reported on the lower part of the Table 5. While the Phone Obsession (PO) Factor appears to be very sensitive to all the model factors with the only exception of the Number of SNS, even if with effects always moderated ranging between 2% and 5%, the Communication Disturbance (CD) Factor shows a sensitivity only towards ICT Pervasivity, Number of SNS, SMS, Social Media, and Internet usage addiction scales. The psychological features of the model seem to affect only the PO Factor.

**Table 5.** Multivariate model.

| MANCOVA General Model ($r^2$: 0.36) | | | | |
|---|---|---|---|---|
| Factor | Wilks' $\lambda$ | F | Power ($\beta$) | $\eta^2$ |
| ICT Pervasitvity | 0.969 | 5.616(2, 351) *** | 0.857 | 3.1% |
| Number of SNSs | 0.969 | 5.639(2, 351) *** | 0.859 | 3.1% |
| SMS usage | 0.946 | 10.114(2, 351) *** | 0.985 | 5.4% |
| Social media usage | 0.928 | 13.639(2, 351) *** | 0.998 | 7.2% |
| Internet usage | 0.941 | 10.961(2, 351) *** | 0.991 | 5.9% |
| Neuroticism | 0.977 | 4.211(2, 351) ** | 0.737 | 2.3% |
| STAI (Trait) | 0.969 | 5.620(2, 351) *** | 0.857 | 3.1% |
| Virtual Sense of Community | 0.977 | 4.160(2, 351) ** | 0.732 | 2.3% |

| Principal effects and Parameters | | | | | |
|---|---|---|---|---|---|
| Parameter | Phubbing Factor | F(Df) | $\beta$ | Student $t$ | Power ($\beta$) | $\eta^2$ |
| ICT Pervasivity | Phubbing $^{CD}$ | 9.430(1) *** | 0.307 | 3.071 *** | 0.865 | 2.6% |
| | Phubbing $^{PO}$ | 5.613(1) * | 0.266 | 2.369 * | 0.656 | 1.6% |
| Number of SNSs | Phubbing $^{CD}$ | 3.820(1) * | −0.210 | −1.954 * | 0.496 | 1.5% |
| | Phubbing $^{PO}$ | ns | - | - | - | - |
| SMS usage addiction | Phubbing $^{CD}$ | 7.473(1) *** | 0.116 | 2.734 *** | 0.778 | 2.1% |
| | Phubbing $^{PO}$ | 18.696(1) *** | 0.207 | 4.324 *** | 0.991 | 5% |
| Social Media usage addiction | Phubbing $^{CD}$ | 18.902(1) *** | 0.106 | 4.348 *** | 0.991 | 5.1% |
| | Phubbing $^{PO}$ | 18.303(1) *** | 0.118 | 4.278 *** | 0.989 | 4.9% |
| Internet usage addiction | Phubbing $^{CD}$ | 20.147(1) *** | 0.158 | 4.489 *** | 0.994 | 5.4% |
| | Phubbing $^{PO}$ | 8.299(1) *** | 0.114 | 2.881 *** | 0.819 | 2.3% |
| Neuroticism | Phubbing $^{CD}$ | ns | - | - | - | - |
| | Phubbing $^{PO}$ | 8.179(1) *** | 0.263 | 2.861 *** | 0.814 | 2.3% |
| STAI (Trait) | Phubbing $^{CD}$ | ns | - | - | - | - |
| | Phubbing $^{PO}$ | 9.531(1) *** | −0.060 | −3.087 *** | 0.868 | 2.6% |
| Virtual Sense of Community | Phubbing $^{CD}$ | ns | - | - | - | - |
| | Phubbing $^{PO}$ | 7.881(1) *** | −0.059 | −2.808 *** | 0.800 | 2.2% |

*** = $p$ 0.001, ** = $p$ 0.01, * = $p$ 0.05. $^{CD}$—Phubbing factor: Communication disturbance; $^{PO}$—Phubbing factor: Phone Obsession; ns: not significant.

## 7. Discussion

This study was driven by the desire to better understand the phubbing phenomenon that is still new and, despite all, not exhaustively investigated. Overall, our paper provides a multidimensional model of phubbing. Having a clear set of factors related with phubbing, will be useful for all those web platforms and online services that can be reached via mobile devices. Indeed, it would be possible for them to evaluate users' phubbing risk by means of a pretty economic profiling phase.

Phubbing does not appear to be exclusively related to addiction behaviors. Nevertheless, our results highlighted a strong connection of phubbing with online addiction behaviors (e.g., Social media addiction, Internet addiction) as well as with psychological and psychosocial determinants of online compulsive behaviors (i.e., Trait and Social Anxiety). Our findings appear in line with the previous literature that defined phubbing as a compulsive behavior put in place to reduce anxiety and discomfort due to social interactions [14,17]. For instance, we reported a positive correlation between phubbing and both trait anxiety and social anxiety, confirming that those with a higher level of anxiety are those who do more phubbing. However, our results suggest that phubbing could be related also with constructs not directly linked to addiction behaviors (e.g., ICT Pervasivity, Virtual Sense of Community). Indeed, when addiction variables were already considered within the multivariate model, other factors still contributed to explain phubbing's variance. Mobile devices seem to be "habit-forming" but these new habits (e.g., checking habit) do not imply necessarily an addiction. Repetitively inspect the content accessible through smartphones could be experienced more as a diversion (sometimes even as an annoyance) than an addiction [42]. Thus, phubbing appears to be a complex phenomenon not only definable and predictable by its addiction component.

Interestingly, a greater number of social networks seemed to push individuals to interrupt face-to-face interaction to check their phones less often. They also experience a lesser need to check

their phones. Probably, those individuals who have many social networks do not give great importance to them, or they could already have a more structured self. In either case, they may not need to interrupt the conversation to check their phone so often.

Moreover, Virtual Sense of Community seem to act as protective factor in our model reducing the phone obsession component. This result may suggest how the successful development of a social identity through virtual environments could reduce the perceived need to engage in phubbing. When addiction related observables were considered within our model, Trait Anxiety also appeared to reduce the phone obsession component of phubbing, while Neuroticism seemed to increase it.

ICTs' availability (i.e., pervasivity) increases the overall phubbing frequency. Usually, the more people are exposed to ICTs and online services in their daily activities, the more their digital media literacy rises [43]. Thus, it is not a surprise that the individuals more used to ICT could also be the ones that more often use smartphones and engage in phubbing. Whether their use can be read as an addition or a simple interruption of face-to-face communication, the "confidence" with ICTs appear to be a promoting factor for phubbing.

We also registered how digital native subjects (i.e., individuals under the age of 26) were more likely to engage in phubbing than digital immigrants. Younger individuals were born and raised in the age of new ICTs [44]. Their use of smartphones is quite different from people who experience ICT revolution in adulthood. Indeed, digital natives' use of smartphones is pervasive in their lives. Not only their use is more frequent with obvious effects on both phubbing components, but it is also more socially connoted. Indeed, digital native individuals use smartphones also to signal their social affiliation as well as to build social relationships [45,46], while digital immigrants, especially seniors, use smartphones mainly for their utility as phones [47]. Therefore, the difference between digital natives and digital immigrants in ICTs' social importance and pervasivity could be a possible explanation for age effect on phubbing.

The social environment defined by dyadic-couple interactions also seemed to have a role in shaping phubbing behavior. Indeed, individual phubbing is associated with the perceived level of a partner's phubbing, which could mean that seeing a significant other engage in phubbing could influence the acceptability of such behavior and possibly reinforce phubbing dynamics [16]. Significant others' influence affects a broad variety of behaviors, among which addiction conducts [48]. Indeed, what others do (i.e., empirical expectations) influences the likely to engage in a behavior, thus defining a norm within a social system [49]. However, our measurement of partners' phubbing is based on a person's perception of what another person does and thus could be biased. Future works should investigate the possible role of phubbing as a shared and self-reinforcing norm. Moreover, future research should also consider the relationship between phubbing and other constructs, for instance selfishness, to better understand phubbing underlying motivations.

Lastly, we verified the effects of gender and psychological and psychosocial observables on phubbing. Differently from the previous literature [11,12], gender did not appear eeither to impact the phubbing level or mediate the relationship between the smartphone addiction, internet addiction and phubbing behavior. The relationship between self-efficacy and anxiety is well known in the scientific literature [20]. Nevertheless, the relation between self-efficacy and phubbing appears quite weak, probably indicating a more complex dynamic between the observables than a simple linear relation.

## 8. Conclusions

Overall, our work could be exploited in line with the primary prevention approach [50]. Indeed, future online services should aim at avoiding the emergence of psychological detrimental issues like phubbing and therefore promoting well-being among Internet users. On the one hand, having a clearer picture of phubbing antecedents could be very useful in assessing users' phubbing potential risk and consequently adapt service-user communication via smartphones. On the other, our results could be used for dedicated mobile device settings to help reduce ICTs' pervasivity for potential phubbers (e.g., managing appropriately the number and the timing of smartphone notifications) [51,52].

Furthermore, given the recent technological advancements in the analysis of heterogeneous data [53], our work could also provide some useful insights about how to use such data to classify users according to their phubbing behavior [54].

**Author Contributions:** Conceptualization, A.G. and P.M.; Data curation, A.C.; Formal analysis, A.G.; Investigation, A.C.; Project administration, A.G.; Resources, P.M.; Supervision, A.C. and P.M.; Writing—original draft, M.D. and A.C.; Writing—review & editing, M.D. and P.M.

**Funding:** This research received no external funding.

**Conflicts of Interest:** The authors declare no conflict of interest.

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
