# Peer review of "An Explorative Model to Assess Individuals’ Phubbing Risk"

_futureinternet, doi:10.3390/fi11010021_

Round 1

Reviewer 1 Report

This is a very interesting work exploring the determinants of phubbing behavior. I only have a set of minor comments.

Abstract

What does ICT mean? Please don’t use undefined acronyms in the abstract (and in the highlights)

Introduction

Lines 53-55. I think that Phubbing could also be related to selfishness: a person ignores the other person to look at their own online image (i.e., prefers the self over the other). I think this possibility also should be mentioned. This makes me think that, since males are more selfish than females (Rand et al, 2016; Branas-Garza et al, 2018), it is possible that, as a consequence, males phub more than females. More generally, potential gender differences in phubbing are worth looking at and should be reported in the results section.

Sampling and participants

Line 94: replace DS with SD

Methods and procedure

Line 101: individuals

I noted that the authors sometimes use “italian” with the lower case “i”. This is incorrect. Please use Italian everywhere.

Data analysis

line 145: skewness (same in line 165, in Table 1, and everywhere else). Also the word average is spelled incorrectly throughout the manuscript (eg., line 175 and 176)

Results

Please add the gender effect

Table 3: it’s not clear whether the term −0.038 is significant or not 

Suggestion for further research:

It would be nice to see whether there is a correlation between phubbing and selfishness, perhaps by correlating the Phubbing scale with the dictator game

References

Branas-Garza P, Capraro V, Rascón-Ramírez E (2018) Gender differences in altruism on Mechanical Turk: Expectations and actual behavior. Economics Letters, 170, 19-23.

Rand DG, Brescoll VL, Everett JAC, Capraro V, Barcelo H (2016) Social heuristics and social roles: Intuition favors altruism for women but not for men. Journal of Experimental Psychology: General, 145, 389-396.

Author Response

#Reviewer 1

Reviewer comment: Abstract What does ICT mean? Please don’t use undefined acronyms in the abstract (and in the highlights).
Author’s answer: Thank you for your comment. We were not completely clear about ICTs acronym. We specified in the abstract and in the highlights section that ICTs stands for Information and Communication Technologies.

Reviewer comment: Introduction. Lines 53-55. I think that Phubbing could also be related to selfishness: a person ignores the other person to look at their own online image (i.e., prefers the self over the other). I think this possibility also should be mentioned. This makes me think that, since males are more selfish than females (Rand et al, 2016; Branas-Garza et al, 2018), it is possible that, as a consequence, males phub more than females. More generally, potential gender differences in phubbing are worth looking at and should be reported in the results section.

Author’s answer: Thank you very much once more. We modified our introduction (Lines 64-67) considering your comment about selfishness and gender-related differences. Moreover, we cited the two works that you kindly provided.

“Phubbing could also be related to selfishness. In other words, phubbers could prefer their own online selves over the social interaction with another individual. Since, males are reported in the scientific literature as more selfish than females (Rand et al, 2016; Branas-Garza et al, 2018), gender could affect phubbing dynamics.

Finally, we added a paragraph in the results section (Lines 250-254) in which we assess potential gender differences:

“Before proceeding to our inferential analyses, we tested by means of a univariate analysis of variance whether gender could affect phubbing. In other words, gender has been introduced as a dummy factor. No statistically significant difference emerged in relation to our dependent variables (i.e., Phubbing Factor: Communication Disturbance, Phubbing Factor: Phone Obsession, Personal Phubbing (Total), Partner Phubbing).   

Reviewer comment: Sampling and participants. Line 94: replace DS with SD

Author’s answer: We replaced DS with SD.

Reviewer comment: Methods and procedure. Line 101: individuals. I noted that the authors sometimes use “italian” with the lower case “i”. This is incorrect. Please use Italian everywhere.

Author’s answer: We replaced “Individuals” with “people” at Line 101. Moreover, we corrected the “Italian” word with the lower case “i” throughout the paper.

Reviewer comment: Data analysis. line 145: skewness (same in line 165, in Table 1, and everywhere else). Also the word average is spelled incorrectly throughout the manuscript (eg., line 175 and 176).

Author’s answer: We corrected the two words that were spelled incorrectly throughout the paper.

Reviewer comment: Results. Please add the gender effect. Table 3: it’s not clear whether the term −0.038 is significant or not.

Author’s answer: The analysis of potential gender effects, as pointed out in the previous comments, is presented in Lines 250-254. We also specify better which results are statistically significant in Table 3 and which are not.

Reviewer comment: Suggestion for further research: It would be nice to see whether there is a correlation between phubbing and selfishness, perhaps by correlating the Phubbing scale with the dictator game

Author’s answer: We agree with your suggestion and we added this paragraph in the discussion section (Lines 349-351):

“Moreover, future research should consider the relationship between Phubbing and other constructs, for instance selfishness, to better understand Phubbing underlying motivations”.

Reviewer 2 Report

The authors propose a multidimensional model for Phubbing considering also behaviors and habits.

The proposed approach is generally well written and presented but in some points does not provide enough details about Methods and Procedures section. In particular, the authors should provide more details about the used scale for addiction instead of citing them.

Moreover, I suggest to cite the following papers to underline how technological advancements enrich the analysis with heterogeneous data and, on other hand, to analyze  how it is possible to use them for classifying users based on their behaviors:

1) Benchmarking big data architectures for social networks data processing using public cloud platforms. Future Generation Computer Systems89, 98-109.

2) Centrality in heterogeneous social networks for lurkers detection: An approach based on hypergraphs. Concurrency and Computation: Practice and Experience, 30(3), e4188.

Finally, a linguistic revision is necessary.

Author Response

#Reviewer 2

Reviewer comment: The proposed approach is generally well written and presented but in some points does not provide enough details about Methods and Procedures section. In particular, the authors should provide more details about the used scale for addiction instead of citing them.

Author’s answer: Thank you for your suggestion. We deeply revised the methods and procedures section. For each scale used in our study we provide further information like the number of items, the type of measurement (e.g., 5-points Likert scale), the meaning of the extreme values of the scale, the reliability of the scale, its dimensionality and we also added some item examples.

Reviewer comment: Moreover, I suggest to cite the following papers to underline how technological advancements enrich the analysis with heterogeneous data and, on other hand, to analyze  how it is possible to use them for classifying users based on their behaviors:

1) Benchmarking big data architectures for social networks data processing using public cloud platforms. Future Generation Computer Systems, 89, 98-109.

2) Centrality in heterogeneous social networks for lurkers detection: An approach based on hypergraphs. Concurrency and Computation: Practice and Experience, 30(3), e4188.

Author’s answer: Thank you very much for your comment. We added in the discussion section a paragraph regarding the analysis with heterogeneous data (Lines 365-368).

“Furthermore, given the recent technological advancements in the analysis of heterogeneous data, our work could also provide some useful insights about how to use such data to classify users according to their Phubbing behavior”.

Moreover, we cited the two works that you kindly suggested.